Corticospinal excitability remains unchanged in the presence of residual force enhancement and does not contribute to increased torque production

Frischholz Jasmin 1
Raiteri Brent J. 1
Cresswell Andrew G. 2
Hahn Daniel daniel.hahn@rub.de 1 2
1 Human Movement Science, Faculty of Sport Science, Ruhr University Bochum , Bochum , Germany
2 School of Human Movement and Nutrition Sciences, University of Queensland , Brisbane , Australia
Cè Emiliano
Electronic publication date: 2022 Jan 6
Publication date: 2022
Volume: 10
Electronic Location ID: e12729
Received 2021 Apr 13; Accepted 2021 Dec 10
Copyright: ©2022 Frischholz et al.
Copyright year: 2022
Copyright holder: Frischholz et al.
License: This is an open access article distributed under the terms of the Creative Commons Attribution License, which permits unrestricted use, distribution, reproduction and adaptation in any medium and for any purpose provided that it is properly attributed. For attribution, the original author(s), title, publication source (PeerJ) and either DOI or URL of the article must be cited.
License URL: https://creativecommons.org/licenses/by/4.0/

Keywords: Eccentric contraction, Active stretch, Neural control, Transcranial magnetic stimulation, Inhibition, Torque enhancement, History dependence

Funding: The authors received no funding for this work.

==============================
Background

Following stretch of an active muscle, muscle force is enhanced, which is known as residual force enhancement (rFE). As earlier studies found apparent corticospinal excitability modulations in the presence of rFE, this study aimed to test whether corticospinal excitability modulations contribute to rFE.

Methods

Fourteen participants performed submaximal plantar flexion stretch-hold and fixed-end contractions at 30% of their maximal voluntary soleus muscle activity in a dynamometer. During the steady state of the contractions, participants either received subthreshold or suprathreshold transcranial magnetic stimulation (TMS) of their motor cortex, while triceps surae muscle responses to stimulation were obtained via electromyography (EMG), and net ankle joint torque was recorded. B-mode ultrasound imaging was used to confirm muscle fascicle stretch during stretch-hold contractions in a subset of participants.

Results

Following stretch of the plantar flexors, an average rFE of 7% and 11% was observed for contractions with subthreshold and suprathreshold TMS, respectively. 41–46 ms following subthreshold TMS, triceps surae muscle activity was suppressed by 19–25%, but suppression was not significantly different between stretch-hold and fixed-end contractions. Similarly, the reduction in plantar flexion torque following subthreshold TMS was not significantly different between contraction conditions. Motor evoked potentials, silent periods and superimposed twitches following suprathreshold TMS were also not significantly different between contraction conditions.

Discussion

As TMS of the motor cortex did not result in any differences between stretch-hold and fixed-end contractions, we conclude that rFE is not linked to changes in corticospinal excitability.

Introduction

It is well known that stretch of an active muscle results in increased force production during the isometric steady state following stretch compared with the steady-state force produced at the same muscle length and activation level during a fixed-end contraction. This is referred to as residual force enhancement (rFE), which was initially investigated in situ (Abbott & Aubert, 1952). Decades later, rFE was investigated in isolated muscle fibres and two opposing mechanisms were suggested: (1) that rFE following active muscle stretch results from non-uniformities in sarcomere lengths (Julian & Morgan, 1979) , and (2) that rFE is due to the engagement of a parallel non-contractile element during active stretch (Edman, Elzinga & Noble, 1978). Since then, numerous studies have investigated the development of rFE and suggested additional potential underlying mechanisms. These suggestions include stretch-induced increases in the number of cross-bridge attachments and/or the attachment of the second myosin head, and an increase in the average cross-bridge force and strain (Brunello et al., 2007; Rassier, 2012; Herzog, 2014). Lately, the engagement of a parallel non-contractile element during active stretch has been related to titin, which might increase its force contribution in the presence of calcium and by interacting with actin (Nishikawa, 2016; Herzog, 2018; Freundt & Linke, 2019).

Complementary to in vitro research, several in vivo studies have investigated rFE (as approximated by net joint torque) during electrically-stimulated and voluntary contractions (Seiberl, Power & Hahn, 2015; Chapman et al., 2018). rFE during voluntary contractions has been observed for small muscles of the thumb (Lee and Herzog, 2002), for large muscles of the lower limb (Pinniger & Cresswell, 2007; Seiberl et al., 2010), and during multi-joint multi-muscle contractions of the lower limbs (Hahn et al., 2010). The potential relevance of rFE during human movement was demonstrated in studies that investigated rFE during submaximal voluntary contractions and at joint angle configurations that mimicked those of human locomotion (Hahn et al., 2010; Seiberl et al., 2013; Paternoster et al., 2016).

Besides rFE, several in vivo studies have documented an activation reduction (AR) during the steady state following active muscle stretch. In contrast to rFE, which is observed when muscle activity is matched between stretch-hold (STR) and fixed-end reference (REF) contractions, AR occurs when force or torque is matched between contraction conditions. AR refers to a reduced muscle activity level needed to maintain a given force/torque following active muscle stretch compared with fixed-end contractions at the same final muscle length. Several studies have reported AR and concluded that neuromuscular efficiency following active muscle stretch is increased, which might be explained by enhanced passive forces due to increased titin forces (Oskouei & Herzog, 2005; Altenburg et al., 2008; Seiberl et al., 2012; Joumaa & Herzog, 2013; Jones, Power & Herzog, 2016; Mazara et al., 2018; Paquin & Power, 2018). However, Paquin & Power (2018) also found a rightward shift in the EMG-torque relationship following active stretch compared with fixed-end reference conditions, which indicates that the neuromuscular activation strategy might be altered in the presence of rFE.

Changes in EMG following active stretch have motivated a number of studies to investigate the neural control and/or neural modulations that occur during the isometric steady state following active muscle stretch (Altenburg et al., 2008; Hahn et al., 2012; Paquin & Power, 2018; Sypkes et al., 2018; Contento, Dalton & Power, 2019; Jakobi et al., 2020). For example, Altenburg et al. (2008) examined single motor unit behaviour of the vastus lateralis muscle during AR following active muscle stretch. These authors found similar discharge rates of VL motor units between STR and REF conditions, which led them to conclude that a derecruitment of motor units might have occurred during AR (Altenburg et al., 2008).

Two further studies used motor evoked potentials (MEPs) and cervicomedullary motor evoked potentials (CMEPs) to investigate excitability modulations at cortical and spinal sites in the presence of rFE (Hahn et al., 2012; Sypkes et al., 2018). In the first study, Hahn et al. (2012) found larger MEPs and larger V-waves, but unchanged CMEPs, following stretch of the plantar flexors during maximal voluntary contractions. Based on the larger MEPs and unchanged CMEPs, all normalised to Mmax, it was interpreted that cortical excitability increased, but spinal excitability was unchanged in the rFE state. The increased V-waves were considered to represent greater cortical motoneuronal output and/or an increase in spinal stretch reflex excitability following active muscle stretch (Hahn et al. (2012). In the second study, Sypkes et al. (2018) found smaller CMEPs (normalised to Mmax) and an unchanged MEP/CMEP ratio in the presence of rFE during submaximal dorsiflexion contractions, which was interpreted as reduced spinal excitability, but no change in cortical excitability. However, because it is not reported whether the MEP/Mmax ratio changed or remained constant, and Mmax varied within participants from −20 to 17% between the ISO and RFE conditions in that study, it is virtually impossible to interpret the provided MEP/CMEP ratio (see supplementary material Table 1 for a detailed explanation). Further, assuming that the MEPs’ corresponding Mmax was unchanged, the combined results of the reduced CMEPs and unchanged MEPs in the study of Sypkes et al. (2018) can also be interpreted as reduced spinal excitability and increased cortical excitability (Martin, Gandevia & Taylor, 2006; Martin et al., 2009). Accordingly, both studies on neural excitability (Hahn et al., 2012; Sypkes et al., 2018) might indicate increased cortical excitability in the presence of rFE, but this currently remains a matter of debate.

Despite different muscle groups being tested under different levels of voluntary effort, the ambiguous results from the earlier studies likely rise due to both studies being underpowered and the statistical significance occurring due to chance. That is, the observed changes in CMEPs (Sypkes et al., 2018) and MEPs (Hahn et al., 2012) in the rFE state, while significant, were not large (Cohen’s dz = 0.70 and 0.75, respectively). A paired t-test with 11 participants, which was the maximum sample size tested across these studies, would not have been able to reliably detect effects smaller than a Cohen’s dz of 0.94 with 80% power at a two-tailed alpha level of 0.05. Further, only a few responses (i.e., MEPs) to transcranial magnetic stimulation (TMS) were averaged (n = 6 in Hahn et al., 2012 and n = 4 minus outliers in Sypkes et al., 2018), although it has been shown that as many as 20 MEPs are needed to accurately estimate corticospinal excitability (Brownstein et al., 2018).

Table 1 EMG responses following subthreshold transcranial magnetic stimulation.

Mean ± SD values of EMG amplitude suppression following subthreshold transcranial magnetic stimulation from soleus (SOL), medial gastrocnemius (MG) and lateral gastrocnemius (LG) muscles during stretch-hold (STR) and fixed-end reference (REF) contractions.

 	SOL	MG	LG	
EMG suppression REF [%]	24.0 ± 6.7	24.7 ± 7.2	19.4 ± 2.3	
EMG suppression STR [%]	24.2 ± 5.4	23.7 ± 6.9	20.9 ± 6.4	
Latency REF [ms]	44 ± 4	41 ± 5	44 ± 5	
Latency STR [ms]	45 ± 3	43 ± 6	46 ± 5	
Duration REF [ms]	14 ± 4	15 ± 4	12 ± 5	
Duration STR [ms]	13 ± 6	12 ± 4	10 ± 4	

Therefore, the aim of this study was twofold. First, we wanted to partly replicate the above-mentioned studies and investigate whether corticospinal excitability was altered in the presence of rFE. This was done by eliciting 20 MEPs via TMS in the presence (STR) and absence (REF) of rFE in an adequately-powered study. Second, this study was designed to investigate whether cortical inhibition induced by subthreshold TMS affects force production in the presence of rFE differently to fixed-end reference contractions without rFE.

Based on a critical evaluation of previous research (Hahn et al., 2012; Sypkes et al., 2018), we predicted that the activation of cortical interneurons and pyramidal neurons by suprathreshold TMS (Rothwell, 1997; Rossini et al., 2015) would result in similar MEPs and superimposed twitches in the presence of rFE compared with the fixed-end reference contractions with matched background muscle activity. Further, we predicted that inhibiting motor cortical neurons by subthreshold TMS (Davey et al., 1994; Petersen et al., 2001; Zuur et al., 2010) would lead to a similar suppression in muscle activity (EMG) and plantar flexion torque in the presence of rFE compared with fixed-end reference contractions with matched background muscle activity. If both predictions hold, the results would indicate that corticospinal excitability is not altered and does not contribute to the increased torque production in the presence of rFE.

Materials & Methods

Participants

Fourteen recreationally-active participants (six women, 26.7 ± 5.3 yrs., 1.77 ± 0.11 m, and 74.0 ± 16.8 kg) voluntarily participated in this study after providing free written informed consent. A total sample size of 10 was calculated to have over 90% power to detect a minimum effect size of 0.62 with a two-tailed alpha level of 0.05. This was calculated with Superpower (https://shiny.ieis.tue.nl/anova_power/) from 2000 simulations using data from (Hahn et al., 2012), which incorporated a 2 ×2 repeated-measures design and observed a common standard deviation of 5.11 (note a conservative within-subjects factor correlation of 0.5 was used in the simulations). Participants were free of any neuromuscular disorders and injuries to their right lower limb. The experimental protocol was approved by the local Ethics Committee of the Faculty of Sport Science at Ruhr University Bochum, Germany (EKS10072018).

Experimental setup

During the experiment, participants laid prone on a dynamometer bench with their upper body supported by their forearms and their right foot tightly strapped onto the footplate attachment of a dynamometer (IsoMed 2000, D&R Ferstl GmbH, GER). The axes of rotation of the ankle joint and the dynamometer were aligned prior to testing and the foot was firmly secured to the footplate using three straps (forefoot, ankle, heel) to minimize heel lift during the plantar flexion contractions. Additionally, the participants’ hips were secured to the dynamometer bench with a belt. A computer monitor positioned directly in front of the participants was used to provide visual feedback of their soleus muscle activity (moving 0.25-s root-mean-square amplitude calculation) during the plantar flexion contractions. Net ankle joint torques and ankle joint angles of the right leg were sampled at 1 kHz using a 16-bit Power-3 1401 and Spike2 data collection system (v.8.01, Cambridge Electronics Design, Cambridge, UK).

Surface electromyography

Muscle activity and responses to TMS from soleus (SOL), medial gastrocnemius (MG) and lateral gastrocnemius (LG) muscles were recorded via bipolar surface electromyography (EMG). EMG of the antagonistic tibialis anterior (TA) was not obtained as it has been shown that TA EMG during fixed-end plantar flexion is more likely from crosstalk than coactivation (Raiteri, Cresswell & Lichtwark, 2015). After skin preparation, self-adhesive surface electrodes (Ag/ AgCl, Kendall, ECG Electrodes, eight mm recording diameter) were positioned with an inter-electrode distance of 20 mm over the triceps surae muscle bellies following the recommendations of SENIAM (Hermens et al., 1999). A reference electrode was placed over the right fibular head. Cables were fixed to the skin with tape to prevent motion artefacts. EMG signals were amplified 1000 times by a multichannel analogue amplifier (AnEMG12, Bioelectronica, Turin, IT) and band-pass filtered between 10 Hz and 4.4 kHz, prior to being sampled at 5 kHz. EMG and torque/angle data were synchronised and sampled using the same analogue-to-digital converter and software described above.

Transcranial Magnetic Stimulation (TMS)

TMS (MagPro Compact, MagVenture, Farum, Denmark) was used to either inhibit or activate the motor cortical area of the right plantar flexor muscles in the left hemisphere, slightly left of the vertex, via a double-cone coil (D-B80 Butterfly Coil, MagVenture, Farum, Denmark). The coil had monophasic current running through its centre in an anterior-posterior direction. The vertex was marked on the scalp and defined as halfway between the left and right processus zygomaticus ossis temporalis and halfway between the os nasale and the external occipital protuberance. The vertex location helped to find the optimal location for TMS of the motor cortical area innervating the right plantar flexor muscles (TMS hotspot), which is generally defined as the position in which a single stimulation evokes the largest peak-to-peak MEP amplitude in the target muscle (Siebner and Ziemann, 2007) . In order to find the TMS hotspot, several stimuli were delivered while the TMS coil was slightly left, in front or behind the vertex, while participants performed fixed-end plantar flexion contractions at 30% of their maximal voluntary EMG activity (MVA) as measured during maximal voluntary fixed-end contractions (see experimental protocol). Once the TMS hotspot was located, the wings of the coil were marked on the scalp with a semi-permanent marker. Subthreshold and suprathreshold intensities were then determined by either decreasing or increasing the stimulator output. To achieve a suppression of soleus muscle activity, stimulator output was reduced in small increments until the active motor threshold (AMT) was reached. This threshold was defined as the point when stimulation during 30% MVA resulted in visible MEP responses in only five of ten consecutive trials (Petersen et al., 2001). Once the AMT was determined, stimulation intensity was slightly reduced again so that MEPs were no longer elicited. For activation of the motor cortex, stimulator output was increased until MEPs were clearly visible in comparison to the immediately preceding background EMG in at least five consecutive trials.

Ultrasound imaging

Muscle fascicle behaviour of the MG from three participants was examined during familiarisation sessions using B-mode ultrasound imaging (LS128 CEXT-1Z, Telemed, Vilnius, Lithuania) to ensure muscle fascicle stretch occurred during ankle rotation in the stretch-hold condition. Ultrasound images were recorded using a flat, linear, 128-element transducer (LV7.5/60/128Z-2, 6 MHz, 60 ×50 mm (width ×depth)) operating at ∼60 fps in EchoWave II software (Telemed, Vilnius, Lithuania). The transducer was placed on the skin over the MG mid-muscle belly in the longitudinal plane and rotated to obtain a clear image with continuous aponeuroses and muscle fascicles. The position of the probe was marked on the skin to ensure consistent placement during the stretch-hold contractions.

Contraction conditions

The experiment involved participants performing submaximal plantar flexion contractions at 30% MVA. The background activity level was controlled throughout the contractions by having participants visually match their SOL EMG amplitude (moving 0.25-s root-mean-square (RMS) amplitude calculation) to a horizontal cursor on a computer monitor. The conditions involved fixed-end reference contractions (REF) at 20° dorsiflexion (DF) and active stretch-hold contractions (STR) from 0° −20° DF (0° refers to the sole of the foot being perpendicular to the shank).

All contractions started with a 2-s linear ramp from 0–30% MVA. Following the ramp, the isometric steady state during REF was maintained for 13-s. During STR, the ramp was followed by a 1-s isometric steady state at the initial ankle joint angle (0° DF), before the active plantar flexor muscles were stretched to an ankle joint angle of 20° DF at an angular velocity of 30° s−1. Following active stretch, participants maintained the subsequent isometric steady state for ∼11-s, resulting in an overall contraction duration of 15-s (Fig. 1).

Figure 1 Example data from the stretch-hold (STR, blue) and fixed-end reference (REF, grey) contraction conditions.

(A) The traces show soleus (SOL) EMG (moving 0.25-s root-mean-square (RMS) amplitude calculations). Transcranial magnetic stimulation (TMS, vertical black lines) was delivered at 1 Hz from 5 s after contraction onset (marked as time zero). In case the first stimulation was delivered before SOL EMG reached the target level, the stimulation was excluded from analysis. (B) Traces show the corresponding crank arm angles.

Experimental protocol

Participants attended two sessions on two different days. In the first session, participants were familiarised with the test protocol and trained to perform maximal voluntary fixed-end contractions (MVC) and the submaximal contractions described above. Additionally, participants were familiarised with TMS.

The second session consisted of the test protocol (Fig. 2). After a short warm-up (eight submaximal plantar flexion contractions with increasing torque), at least two MVCs were performed to determine 100% MVA. Participants were instructed to push their forefoot into the footplate as hard as possible and to maintain the contractions so that a torque plateau was clearly visible. MVC torque was calculated as the difference between peak torque during the contraction and the mean baseline over 500 ms prior to the beginning of contraction. To ensure that the MVCs were repeatable, peak-to-peak torques were required to be within a 5% range. 100% MVA was determined as the smoothed (moving 0.25-s RMS amplitude calculation) SOL EMG amplitude at peak MVC torque from the MVC with the highest peak-to-peak torque. Three minutes of rest was provided between MVCs to minimise fatigue.

Figure 2 Schematic diagram of the experimental protocol.

(A) The protocol started and ended with maximal voluntary contractions (MVC, dark grey). Transcranial magnetic stimulation (TMS) hotspot and sub-/suprathreshold intensities (light grey) were then determined before data collection. Twenty contractions with subthreshold TMS (dark blue) and 4 contractions with suprathreshold TMS (light blue) were separated into three sets, with 5 min rest between each set. (B) Schematic description of contractions within sets (blue bars) and delivery of TMS (same procedure for sub- and suprathreshold TMS). Vertical black lines indicate the timing of TMS during the submaximal contractions. Each set during subthreshold TMS consisted of 10 contractions of 15 s duration, followed by a 3 min rest.

Following the MVCs, the TMS hotspot and the subthreshold and suprathreshold TMS intensities were determined during sustained fixed-end contractions at 30% MVA. Once hotspot and stimulation intensities were determined, 100 subthreshold stimulations were delivered for each contraction condition to study motor cortex inhibition. For this purpose, contractions from each condition (i.e., STR or REF) were separated into ten sets with ten TMS (1 Hz) during the isometric steady state phase of the contractions at 20° DF starting 5-s following contraction onset (Fig. 1). Contractions were randomised and a minimum of three minutes rest was provided between each set. Activation of the motor cortex was investigated by providing 20 suprathreshold stimulations per contraction condition. Suprathreshold stimulations were delivered at the same time points as the subthreshold stimulations, but participants performed only two contractions per condition. The contractions with suprathreshold TMS were always performed after all sets of the subthreshold stimulations had been completed. During all contractions, trials were excluded and repeated if participants could not maintain their soleus EMG activity within 25–35% MVA.

Data analysis

Residual Force Enhancement (rFE)

rFE for the subthreshold and suprathreshold conditions was determined by calculating the difference between the isometric steady-state torque during the STR and REF conditions. Net plantar flexion torque was averaged across contractions for each contraction condition and mean rFE was calculated during the isometric steady state at 20° DF, from 500–990 ms after each TMS stimulus. While most rFE studies have analysed torque or force with a 500 ms time window, we excluded the final 10 ms to ensure that the stimulations delivered at 1 Hz would not affect our analysis. Stimulations delivered before the isometric steady state of SOL muscle activity were excluded. In cases where participants showed no rFE for a specific stimulation condition (i.e., suprathreshold or subthreshold TMS), their data were excluded from statistical analysis..

TMS inhibition

Suppression of muscle activity (SOL, MG and LG) following subthreshold TMS was determined via averaged rectified raw EMG signals from each muscle and contraction condition. Waveform averages were calculated over a time window of 100 ms after stimulation and again, stimulations delivered before the isometric steady state of SOL muscle activity were excluded. The latency and duration of EMG amplitude suppression were determined using the evaluation methods by Petersen et al. (2001) and Zuur et al. (2010). The onset of EMG amplitude suppression was marked when the EMG amplitude first dropped under the background EMG amplitude (mean value calculated 15–25 ms following stimulation) for at least 4 ms. The offset of EMG amplitude suppression was calculated when the EMG amplitude rose above the background EMG amplitude for at least 1 ms (Zuur et al., 2010) (Fig. 3). The mean EMG amplitude between the onset and offset of suppression was calculated and compared with the background EMG amplitude right before suppression.

Figure 3 Example data of rectified and averaged soleus (SOL) EMG from contractions with subthreshold transcranial magnetic stimulation (TMS).

The horizontal dotted line represents the averaged background EMG amplitude calculated from 15-25 ms following stimulation. Vertical dashed lines indicate onset and offset of EMG amplitude suppression following TMS. Time zero indicates the time of stimulation.

Torque production following TMS-induced EMG amplitude suppression was analysed via moving correlations (50-ms time intervals) between the torque of the STR and REF conditions. A reduction in torque due to TMS-induced EMG amplitude suppression was identified when torque data of REF and STR contractions were correlated (r ≥ 0.7, i.e., a very large effect). This data analysis was based on the random fluctuations in torque steadiness that should result in non-correlated torque signals. However, when torque was affected by the TMS-induced EMG amplitude suppression in a similar manner, large positive correlations were expected. Once the time window of torque reduction was identified, the torque offset between the REF and STR conditions was removed by subtracting the mean torque difference between REF and STR over the first 10 ms after the onset of torque reduction. Finally, the mean torque during the period of TMS-induced torque reduction (i.e., during the times where r ≥ 0.7) was calculated for REF and STR and compared between contraction conditions.

TMS activation

MEPs of SOL, MG and LG following suprathreshold TMS were calculated as peak-to-peak amplitudes from the raw EMG signals and averaged across contractions for each muscle and contraction condition (Lewis et al., 2014) (Fig. 4). Additionally, the silent period (SP) duration was analysed for both contraction conditions as the time from stimulation to the end of the SP. The end of the SP was defined as the time when the EMG signal following the MEP exceeded the threefold standard deviation (SD) of the raw EMG during the visually apparent SP (Fig. 4). SD during the SP was determined as the smallest SD of the raw EMG over a moving 30-ms window. The sizes of superimposed twitch torques following suprathreshold TMS were calculated separately as the difference in torque between the peak of the twitch and the torque at the time of stimulation and averaged for each contraction condition.

Figure 4 Representative example of motor evoked potentials (MEPs) and silent periods (SPs) from raw soleus (SOL) EMG signals elicited by suprathreshold transcranial magnetic stimulation (TMS).

The light grey traces show the single MEPs and the dark blue trace shows the average of all SOL MEPs from one participant. The horizontal dashed black line indicates the duration of the silent period (SP) and the vertical black line indicates the peak-to-peak amplitude of the largest single MEP. Time zero indicates the time of stimulation.

Statistical analysis

A two-way repeated-measures ANOVA was used to assess if the difference in triceps surae muscle activity between REF and STR conditions across the tested muscles were significantly different (contraction type × muscle). Due to missing values, two-way repeated-measures restricted maximum likelihood mixed-effects models were used to test for significant differences in EMG amplitude suppression (six missing values), MEP amplitude (two missing values) and SP duration (five missing values) between REF and STR conditions across the tested muscles (contraction type × muscle). If a significant interaction was observed, Bonferroni post-hoc comparisons were performed to determine which muscle significantly differed between REF and STR conditions. Paired t-tests or Wilcoxon signed-rank tests, based on the normality of paired differences as assessed by Shapiro–Wilk tests, were used to test for significant differences in steady-state plantar flexion torque, torque reduction following subthreshold TMS, and superimposed twitch torque following suprathreshold TMS between REF and STR conditions. The alpha level was set at 0.05 and statistical analysis was performed using Prism 9 software (GraphPad, USA). Values are presented as mean ± SD in the text.

Results

Five and four participants were excluded from statistical analysis as they did not show rFE following the subthreshold (n = 9) and suprathreshold (n = 10) stimulations, respectively.

Contraction conditions

During both contraction conditions, participants managed to maintain a constant level of muscle activity (∼30% MVA). EMG of SOL, MG and LG did not significantly differ between STR and REF conditions (P = 0.504) during the isometric steady state of contractions with suprathreshold TMS. However, for subthreshold TMS, while EMG of SOL or LG was not significantly different between conditions (SOL: P > 0.999, LG: P > 0.999), EMG of MG was significantly higher in the REF compared with STR condition (P = 0.001; see Fig. S1). Ultrasound imaging confirmed that the muscle–tendon unit stretch during the 20° dynamometer rotation of the stretch-hold contraction resulted in muscle fascicle stretch of the MG at 30% MVA (n = 3).

Contractions with subthreshold TMS

The isometric steady-state net plantar flexion torque in the STR condition exceeded the time-matched steady-state torque in the REF condition (STR: 105.9 ± 34.5 Nm; REF: 99.0 ± 33.4 Nm, P = 0.004), which resulted in mean rFE of 7.3 ± 4.2% (Fig. 5A).

Figure 5 Residual force enhancement (rFE), motor evoked potentials (MEPs), and twitch torques.

(A) Residual force enhancement during stretch-hold (STR) contractions normalized to the time-matched torque during fixed-end reference (REF) contractions. The open circles and error bars represent the means and 95% confident intervals for the contractions with subthreshold transcranial magnetic stimulation (TMS) (left) and suprathreshold TMS (right), respectively. The grey dots represent the individual data points. (B) Normalised MEP amplitude differences between STR and REF. The open circles and error bars represent the means and the 95% confident intervals and the grey dots represent the individual data points for soleus (SOL), medial gastrocnemius (MG) and lateral gastrocnemius (LG) muscles, respectively. (C) Superimposed twitch torques after suprathreshold TMS. The open circles and error bars represent the means and the 95% confident intervals and the grey dots represent the individual data points for REF (left) and STR (right).

Subthreshold stimulation of the motor cortex led to significant (P < 0.001) EMG amplitude suppression for SOL, MG and LG without any significant difference in amplitude suppression (P = 0.794) between STR and REF conditions. The mean EMG amplitude suppression ranged between 19–25% of the background EMG amplitude, it occurred 41-46 ms after stimulation, and it lasted 10-15 ms in total (Table 1). After accounting for the torque offset between the REF and STR conditions, the net plantar flexor torque following EMG amplitude suppression did not significantly differ (P = 0.729) between STR and REF conditions.

Contractions with suprathreshold TMS

The isometric steady-state net plantar flexion torque in the STR condition exceeded the time-matched steady-state torque in the REF condition (STR: 89.4 ± 26.0 Nm; REF: 82.2 ± 27.2 Nm, P = 0.002), which resulted in mean rFE of 10.8 ± 10.0% (Fig. 5A). For all muscles analysed (SOL, MG, LG), MEP amplitudes (P = 0.529) and SP durations (P = 0.609) following suprathreshold TMS did not significantly differ between STR and REF conditions (MEP: Fig. 5B; SP: Table 2). TMS-evoked superimposed twitches following suprathreshold stimulations were 9.1 ± 3.5 Nm (STR) and 8.6 ± 3.5 Nm (REF), and did not significantly differ between STR and REF conditions (P = 0.131, Fig. 5C).

Discussion

The aims of this study were to determine whether corticospinal excitability is modulated in the presence of rFE and to assess whether an inhibition of motor cortical neurons affects muscle activity and the increased torque production in the presence of rFE compared with fixed-end reference contractions without rFE. To achieve this, we compared reductions in EMG activity and torque production following inhibition of motor cortical neurons, and MEPs, SPs and superimposed twitch torques following activation of motor cortical neurons between submaximal stretch-hold and fixed-end contractions.

Participants managed to keep muscle activity constant (30% MVA) throughout the fixed-end reference contractions and during the isometric steady state of the stretch-hold contractions. As the observed differences in MG background EMG were rather small (2% MVA), they are considered negligible regarding the interpretation of the results. Mean rFE magnitudes of 7% (subthreshold TMS) and 11% (suprathreshold TMS) were observed during the steady state in STR compared with REF, which is in line with former studies (Oskouei & Herzog, 2005; Pinniger & Cresswell, 2007; Seiberl et al., 2013; Paternoster et al., 2016). Inhibition of motor cortical neurons by subthreshold TMS caused reductions in EMG activity and net plantar flexion torque, however the reductions did not differ significantly between contraction conditions. Similarly, MEP amplitudes, SP durations and superimposed twitch torque amplitudes evoked by suprathreshold TMS were not significantly different between STR and REF conditions.

Table 2 EMG responses following suprathreshold transcranial magnetic stimulation.

Mean ±SD values of motor evoked potentials (MEPs) and silent periods (SPs) after suprathreshold transcranial magnetic stimulation from soleus (SOL), medial gastrocnemius (MG) and lateral gastrocnemius (LG) muscles during stretch-hold (STR) and fixed-end reference (REF) contractions.

 	SOL	MG	LG	
MEPs REF [V]	1.30 ± 0.78	2.54 ± 2.67	1.19 ± 0.46	
MEPs STR [V]	1.37 ± 0.72	2.70 ± 2.97	1.14 ± 0.50	
SPs REF [ms]	115 ± 31	112 ± 22	118 ± 29	
SPs STR [ms]	116 ± 31	115 ± 23	123 ± 34	

Following subthreshold TMS, we found a suppression of triceps surae muscle activity by 19–24% relative to the background activity. This is similar to the ∼15% EMG amplitude suppression in SOL induced by subthreshold TMS during walking and jumping (Petersen et al., 2001; Zuur et al., 2010), but smaller compared with the 50% suppression in hand and arm muscles during voluntary fixed-end contractions (Davey et al., 1994). Also, the latency and the duration of the observed EMG amplitude suppression was similar to values reported previously (Petersen et al., 2001; Zuur et al., 2010). Importantly, the TMS-induced EMG amplitude suppression showed no significant difference between STR and REF conditions, which also resulted in similar magnitudes of torque reduction for both contraction conditions. This supports our prediction that inhibiting motor cortical neurons by subthreshold TMS would not affect the STR and REF conditions differently.

Our data also support our prediction that suprathreshold TMS does not elicit larger MEP amplitudes and SP durations, or larger superimposed twitch torque amplitudes in the presence of rFE compared with fixed-end reference contractions (twitch amplitude of ∼9 Nm for both contraction conditions). The unchanged MEP amplitudes and SP durations indicate that corticospinal excitability was unaltered in the presence of rFE, which, based on a re-analysis of the previously published data is in line with the earlier studies on corticospinal excitability in the presence of rFE (Hahn et al., 2012; Sypkes et al., 2018).

Interestingly, we found that activation of the motor cortex with a given suprathreshold stimulation intensity elicited comparable twitch torque amplitudes for both contraction conditions, despite steady-state torques following active stretch being enhanced because of rFE. We think that this finding further supports our interpretation of an unaltered corticospinal in the presence of rFE. This is because increased cortical excitability, but unchanged spinal excitability in the presence or rFE, as observed by Hahn et al. (2012), should lead to larger superimposed twitches, while unchanged cortical excitability, but reduced spinal excitability in the presence of rFE, as observed by Sypkes et al. (2018), should lead to smaller superimposed twitches. However, as neither of these findings were observed in the current study, we think that the overall excitability of the neuromuscular system was unchanged in the presence of rFE.

Finally, from a mechanical point of view, the unchanged superimposed twitches are in opposite to what would be expected based on the work of Merton (1954), who showed that superimposed twitch size decreases as the pre-stimulus torque of the voluntary contraction increases. Accordingly, we interpret the equally-sized superimposed twitches following TMS in the presence of rFE as support for the idea that rFE is not due to a higher number of cross bridges following active stretch, but due to passive structural elements within the muscle that are engaged during active stretch (Edman, Elzinga & Noble, 1978; Nishikawa, 2016; Herzog, 2018; Freundt & Linke, 2019). The contribution of such passive structural elements to the increased force and torque production following stretch could also explain the observed reduction in EMG activity following stretch observed in force/torque-matched contractions (Oskouei & Herzog, 2005; Seiberl et al., 2012).

Limitations

First, we only performed TMS of the motor cortex to assess corticospinal excitability, which does not allow us to distinguish between cortical and spinal aspects of corticospinal excitability. Accordingly, the unchanged MEP amplitudes and SP durations that we found in the presence of rFE compared with the reference contractions do not exclude potential cortical and/or spinal modulations in the presence of rFE. Second, we did not obtain M-waves to normalise MEPs. Although the earlier studies (Hahn et al., 2012; Sypkes et al., 2018) reported statistically unchanged M-wave amplitudes in the presence of rFE, theoretically, we might have missed potential changes in MEP amplitudes due to possible changes in M-wave amplitudes. However, the similar superimposed twitches in the STR and REF conditions provide support that the overall excitability of the neuromuscular system was unchanged. Finally, we did not obtain TMS input–output curves, which would reveal the relative sizes of the measured MEPs. However, when setting up the individual stimulation intensities, we ensured that MEPs could still increase with increasing stimulation intensity so that changes due to the contraction conditions would be detectable.

Conclusions

In conclusion, we found that subthreshold and suprathreshold TMS of motor cortical neurons affected muscle activity and torque production, but the mechanical and neural responses to TMS did not differ between stretch-hold and fixed-end reference contractions. This is in line with our predictions and suggests that corticospinal excitability remains unaltered in the presence of rFE. This further suggests that the enhanced torque production following active muscle stretch is not due to changes in corticospinal excitability, but that rFE is likely caused by a stretch-induced engagement of passive structural elements.

Supplemental Information

Supplemental Information 1 Background EMG during contractions with subthreshold transcranial magnetic stimulation (TMS)

Background EMG amplitudes from the triceps surae muscles following subthreshold TMS for individual participants (blue symbols and grey lines) and the group average (black horizontal lines) during active stretch-hold (STR) and fixed-end reference (REF) contractions.

Click here for additional data file.

Supplemental Information 2 Interpretation of MEPs and CMEPs

Virtual data to demonstrate the effect of altering maximal M-wave (M_max) amplitudes (e.g., due to peripheral excitability changes or configurational changes of the EMG electrodes relative to the underlying muscle fibres) and/or MEP size and/or CMEP size on the MEP/CMEP ratio. In the first virtual test condition, MEPs and CMEPs are reduced by the same amount between reference and test conditions (from 10 to 5), which results in a MEP/CMEP ratio of 1 indicating reduced spinal excitability, but unchanged cortical excitability. However, as the M_max corresponding to the MEPs changed as well (the M-wave MEP became smaller by 20% between reference and test conditions, whereas the M-wave CMEP was unchanged), the normalised MEP/CMEP ratio goes up to 1.25. This virtual finding would subsequently indicate reduced spinal excitability, but increased cortical excitability. In the second virtual test condition, CMEPs are reduced between reference and test conditions (again from 10 to 5), but MEPs are unchanged. While the reduced CMEPs would again indicate reduced spinal excitability, the increased MEP/CMEP ratio of 2 (as expected based on reduced CMEPS, but unchanged MEPs) would indicate increased cortical excitability. However, this calculated ratio might be biased by M-waves varying between conditions. Accordingly, the two test conditions demonstrate that it is virtually impossible to interpret MEP/CMEP ratios when the corresponding M_max changes are unknown.

Click here for additional data file.

Supplemental Information 3 Data for statistical analysis

Click here for additional data file.

Additional Information and Declarations

Competing Interests

Author Contributions

Human Ethics

Data Availability

The authors declare there are no competing interests.

Jasmin Frischholz conceived and designed the experiments, performed the experiments, analyzed the data, prepared figures and/or tables, authored or reviewed drafts of the paper, and approved the final draft.

Brent J Raiteri and Andrew G Cresswell conceived and designed the experiments, analyzed the data, authored or reviewed drafts of the paper, and approved the final draft.

Daniel Hahn conceived and designed the experiments, analyzed the data, prepared figures and/or tables, authored or reviewed drafts of the paper, and approved the final draft.

The following information was supplied relating to ethical approvals (i.e., approving body and any reference numbers):

The local Ethics Committee of the Faculty of Sport Science at Ruhr University Bochum, Germany granted ethical approval to carry out the study within its facility (EKS10072018).

The following information was supplied regarding data availability:

The individual data that was statistically analysed are available in the Supplementary File.

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
