# Peer review of "Corticospinal excitability remains unchanged in the presence of residual force enhancement and does not contribute to increased torque production"

_PeerJ, doi:10.7717/peerj.12729_

## Round 0.1 · original submission · Major Revisions

Dear Authors,

Your review has now been completed for the manuscript "Corticospinal excitability remains unchanged in the presence of residual force enhancement and does not contribute to increased torque production". You will notice that we had three excellent and comprehensive reviews of the work. I believe these comments can help improve the quality of this manuscript and I hope that you consider them carefully.

While the reviewers all agreed that the manuscript was interesting, well conducted and a valuable contribution to the field, you will find some important considerations were raised. In particular, there are concerns about the validity of the reanalysis and the need to shorten/restructure some written sections. Reviewer comments can be found below and I look forward to your reply.

Regards,
Mike

·

Basic reporting

I took the traditional approach and included everything below

Experimental design

I took the traditional approach and included everything below

Validity of the findings

I took the traditional approach and included everything below

Additional comments

Frischholz et al. conducted a nice, comprehensive study looking into corticospinal excitability as related to residual force enhancement. Very nice background and historical context! The work is well presented, and would be a great addition to PeerJ. This group does a fantastic job methodologically to control for muscle fascicle excursions via ultrasound recordings, and the methods/results are presented very clearly. . I present here a few points for the authors to consider:

1) Flexors and extensor muscle groups have differing neural inputs (e.g., incidence of PICs), it’s quite possible that results in the DFers will not match results in the PFers – therefore, differing results to that of Sypkes et al. is not surprising, nor controversial. This point could be raised if warranted. Regarding, “rFE is not linked to changes in CS excitability”, when presenting these statements, consider framing in the context of the muscle group tested. There is considerable data on the history-dependence of force that would indicate CS excitability is altered. Taken together, I would suggest softening any definitive statements throughout the paper not directly related to what the authors measured, or the PFers.

2) “making rFE a muscle mech property rather than a combined neuromuscular property”. Given the body of literature on the HD of force, this statement seems obvious. I would suggest that CNS excitability can be altered by rFE (and FD), and during voluntary contractions, depending on activation level etc, rFE has been shown to also differ in magnitude. There is no doubt rFE is a property of muscle. The quoted statement would make an uninformed reader think this was a new finding. This comment also applies to Line 152-154, and the conclusion. Something like this may be helpful, I’ll leave it up to the authors: “rFE is indeed an intrinsic property of muscle. While our results indicate no alterations to CS excitability, others show that rFE (and rFD) can modulate CS excitability, this may be dependent on muscle group tested, and level of neuromuscular activation”

3) 2nd Aim (Line 139): It’s not clear to me that this aim is trying to achieve. If someone was performing a voluntary contraction during the SP of an evoked MEP, wouldn’t that temporary reduction in force output affect your measures? This aim may need to be clarified.

Line 89: Consider Jakobi et al. (PMID: 32347154 ) to support the single MU statement
Line 337 – did the outliers present with similar CS values?
Line 346 – not matching EMG will affect the evoked CS values, was this accounted for?
The authors considered the limitations of their methods, and presented it very transparently.

Great work! Reviewed by: Geoff Power

·

Basic reporting

With the exception of some minor errors and a small number of specific sentences (noted in the General Comments), the text is well-written and the English will be easily understood by an international audience. Most of the relevant literature is cited and the authors’ rationale for the study is clear. I have no concerns with the quality and labelling of the figures.

Experimental design

While I have serious concerns about the validity of the reanalysis that led to the first aim of the research study (see next section), the intent is clear and the authors explain how both aims address a knowledge gap. The research was conducted carefully and includes the relevant methodological details needed to replicate the experiment. My only concern about experimental design relates to the setting of subthreshold and suprathreshold TMS intensities; i.e., the authors report intensities relative to maximal stimulator output so it would appear there were no criteria based on the actual physiological responses (e.g., percentages of AMT or, in the case of suprathreshold stimulation, a particular MEP amplitude).

Validity of the findings

The raw data have been provided and the interpretations are appropriate, for the most part. I am troubled by the process of including and excluding participants based on the presence of rFE. Looking at the raw data, there were two participants (#2 and 14) who were included in the subthreshold analyses because they displayed rFE across 10 sets of contractions but excluded from the suprathreshold analyses because rFE was not evident with 2 sets of contractions. Conversely, three participants (#8, 10, and 11) were included in the suprathreshold analyses even though they failed to show rFE across the more robust subthreshold protocol. I find both of these scenarios concerning, particularly the second one. For example, in the subthreshold protocol, a participant could show a mean rFE across 10 contractions but fail to display rFE with several combinations of just 2 contractions (meaning, they would be excluded from the suprathreshold protocol if those were the only 2 contractions). Please justify this approach to inclusion and exclusion.

A second point regarding the raw data, there are instances when participants have a value for the STR or REF condition but not both. How did the authors handle these cases when performing statistical analyses? Assuming these participants were excluded, the authors should make it clear in the manuscript how many participants were included for each variable. For example, it would appear that only 6 participants were included in the analysis of MG EMG suppression.

As alluded to in my comment for the previous question, I believe the reanalysed data to be invalid. In the case of reanalysing the data from the 2018 study by Sypkes and colleagues to normalise the MEP to Mmax, the analysis itself is correct but it is unnecessary and the interpretation is incorrect. Normalising the MEP to the CMEP accounts for both spinal and peripheral contributions to the MEP, which means the procedure isolates cortical excitability in each participant. This method to evaluate cortical excitability is far superior to a comparison of mean values of MEP and CMEP normalised to Mmax. As such, the interpretation put forth by Sypkes and colleagues (reduced spinal excitability and unchanged cortical excitability) is accurate and the present authors incorrectly state that those data support increased cortical excitability. The second reanalysis conducted by the authors involved pooling the data from two studies (Hahn et al., 2012 and Sypkes et al., 2018) and is even more problematic than the first reanalysis. Measures of neural excitability (i.e., MEP, CMEP, and, to a lesser extent, Mmax) are highly dependent on the conditions of the task, which makes it inappropriate to pool data collected during maximal contractions (Hahn et al., 2012) with data collected at a moderate intensity (Sypkes et al., 2018). Furthermore, the responses can differ greatly between muscles (e.g., Giesebrecht et al., 2010 vs. Gandevia et al., 1999 or Martin et al., 2006 vs. Oya et al., 2008), which makes it inappropriate to pool data from the soleus (Hahn et al., 2012) and TA (Sypkes et al., 2018). Given these critical issues with the reanalysed data, I do not believe the reanalyses have a place in the manuscript. This will have significant implications on the rationale of the study and would require the elimination of Figure 1.

Additional comments

General note
Although well written, the Introduction is far too long. I consider two double-spaced pages to be typical but the present Introduction is nearly three times that length. Given the issues with the reanalysed data, eliminating all of that text would be a good place to start efforts to shorten the Introduction.

Introduction
- lines 53 & 58: This section of the text refers to isolated muscle fibres so I suggest the authors choose a word other than “recruitment” because of its association with motor units during voluntary contractions.
- lines 86-93: In addition to the Altenburg study, the authors should also report the single motor unit data from the 2020 study by Jakobi et al.
- line 111: Replace “was” with “were”
- lines 124-127: I agree with the authors that it is beneficial to have a large rather than small number of MEPs; however, the cited recommendations relate to responses collected from a relaxed muscle. The variability is considerably greater during relaxation compared to a submaximal contraction (although variability increases again during very strong or maximal contractions), which undermines the relevance of the reported numbers.
- lines 127-135: See earlier comments about the validity of the analysis on the pooled data. Also, the present study has only 9 (subthreshold data) or 10 (suprathreshold data) participants so I would encourage the authors to reconsider an emphasis on the sample size of the previous studies.
- line 129: Has anyone tried to replicate the results? I think there is an important distinction between an inability to replicate findings and a lack of an attempt.
- lines 131-134: What are the units for the CMEPs and MEPs?

Methods
- lines 204-205: It becomes clear later but I suggest the authors make it clear that MVA refers to EMG rather than torque output
- lines 212-216: See my comment about TMS intensities in the Experimental Design section
- lines 231-233: Given that plantar flexor torque reflects the contribution of SOL, MG, and LG, why did the authors use only SOL EMG as feedback to the participants?
- line 257: Replace “peak-to-peak” with “peak”
- line 268: I believe this should refer to Figure 3 rather than 2
- lines 270-272: What was the rationale for only 2 sets of contractions for the suprathreshold protocol?
- lines 273-275: How many trials were excluded? Based on the raw data, it would appear a few of the participants (#1, 3, 7) had significant fatigue development by the time they completed the suprathreshold protocol
- lines 306-307: It is explained a couple of sentences later but it should be made clear here the correlations refer to EMG suppression and torque for each contraction type rather than a correlation between REF and STR torque, which is how it reads at present
- line 307: Replace “was” with “were”
- lines 308-309: Again, make it clear what two variables are being compared with the correlation
- lines 311-313: Why did the authors take this approach to account for the disparity in absolute torque between REF and STR rather than compare the relative decrease in torque for the two contraction types?
- lines 322-323: The end of this sentence is difficult to follow so I suggest rewriting the explanation of determining the end of the SP

Results
- line 341: What do the authors mean by “largely”? Was there some formal (statistical) assessment of their ability to match the target?
- line 372: Delete “your results here”

Discussion
- line 377: If I understand the meaning of the sentence correctly, this should be “decreased” not “increased torque production”
- line 383: See the comment for line 341
- lines 385-386: What is the basis for considering 2% to be a negligible difference? I am not suggesting that is necessarily a meaningful difference but some participants had a mean rFE of ~2% so feel there should be some support for this decision
- line 407: Replace “supports” with “support”
- line 411: As noted in the Limitations section, the MEP is not normalised to the Mmax so “corticospinal” is not technically correct because the MEP reflects excitability of the entire motor pathway
- lines 411-413: See earlier comments about reanalysed data
- lines 417-420: Merton’s data were plotted as absolute forces but the relationship of superimposed twitch size to contraction strength is less about absolute force/torque and more about the activation of the motoneurone pool
- lines 420-423: In light of the previous point and the fact that EMG output was matched for the two contraction types, I question if the superimposed twitch data can be used as they are in this sentence. At a minimum, the speculative nature of this statement needs to be clearer.
- line 429: See the comment for line 411
- lines 431-434: The authors correctly note that spinal modulations cannot be excluded but fail to note that cortical modulations also cannot be excluded
- lines 435-436: Given the earlier comments about the superimposed twitch data, I do not think they can be used to exclude a modulation of spinal excitability
- lines 436-440: Mean Mmax amplitude was not different between REF and STR in the study by Sypkes and colleagues; however, a closer examination of Figure 3 suggests this reflects offsetting increases and decreases in different participants. As such, I think it is still important to normalise MEPs to Mmax in individual participants
- lines 440-442: See earlier comments about the interpretation of the superimposed twitch data
- lines 442-446: While I don’t believe it was necessary to obtain complete TMS input-output curves, the authors are understating the limitation to their approach of “ensuring MEPs could increase with increasing stimulation intensity”. Whether TMS intensity is set relative to a threshold (AMT or RMT) or to obtain a particular MEP size (relative to Mmax to ensure a comparable activation of the motoneurone pool), it is important to have some criterion that is applied in all participants


Tables
- Table 2 legend: To match the legend of Figure 5, make MEP and SP plural
- Table 2: Should MEP amplitudes be in mV rather than V?

Figures
- Figure 5: Should y-axis units be mV rather than V?

Reviewer 3 ·

Basic reporting

no comment

Experimental design

no comment

Validity of the findings

no comment

Additional comments

General Comments
This manuscript provides evidence that residual force enhancement during isometric plantar flexion contractions, is not modulated through corticospinal pathways, and is therefore likely a muscle mechanical property rather than a combined neuro-muscular phenomenon. The manuscript is well written and concise, and provides the reader pertinent information to reach the conclusions stated. However, minor changes are required regarding structure of sentencing that lead to use of the methodology, which are described below:

Abstract:
Line 21: “Following active muscle stretch…..”. The use of the term active muscle stretch in this context, I believe refers to a voluntarily active muscle being passively stretched with a dynamometer. However, when first read it seems a little confusing, and could be perceived as an activation of the muscle to reposition the joint angle voluntarily (i.e. joint angle was changed through voluntary activation of the muscle). Can you please restructure to make this more apparent that an active muscle during the STR condition is passively stretched into position, similar to the opening sentence of introduction.

Methods:
Line 171: Was the computer monitor positioned directly in front of the participants? Please clarify.
Line 183: Were the electrodes placed parallel to the orientation of the muscle fibers for each muscle? Please clarify?
Line 212-216: The reduction and increase of stimulator output following AMT appears to be arbitrary, as currently written. Do the stimulator percentages reported correspond to average stimulator outputs used throughout the experiment for the participant pool?. Please report the percentage that the stimulator was reduced or increased for each participant (i.e. stimulator output was reduced 15% from AMT for subthreshold stimulation, and increased……).
Line 241: “muscle were actively stretched…..”. Would this not be a passive stretching of the muscle by the dynamometer, while maintaining muscle activation?. Please clarify.
Results:
Line 341: “largely managed to maintain….”. Rephrase this statement please. According to your inclusion criteria, EMG was maintained between 25-35%. This sentence makes it appear otherwise.
Discussion:
Line 383: Again please rephrase this sentence. Either the participants maintained contraction levels within the inclusion criteria or they did not.
Limitations
Line 435: This sentence should be removed. The previous sentence outlines that changes may have occurred within a differing structure than outlined in the current experiment. Therefore a conclusion should not be made, or restructured.

---

## Round 0.2 · Minor Revisions

Dear Authors,
two experts in the filed revised your manuscript retrieving some minor issues you should consider within your ms revision process.

·

Basic reporting

please see below

Experimental design

please see below

Validity of the findings

please see below

Additional comments

The authors addressed my initial concerns, and IMHO provided a balanced response to the rest of the reviewers. There are some points we do not see 100% eye to eye on, but given the thoughtful reply I am happy to agree to disagree.
All the best
Geoff Power

·

Basic reporting

No comment.

Experimental design

No comment.

Validity of the findings

No comment.

Additional comments

Thank you to the authors for their detailed responses to my comments and the careful edits to the manuscript, which I view as much improved. Before providing a few comments about the revised manuscript, I will revisit some comments from the original review and the authors’ responses .

Original comment:
As alluded to in my comment for the previous question, I believe the reanalysed data to be invalid. In the case of reanalysing the data from the 2018 study by Sypkes and colleagues to normalise the MEP to Mmax, the analysis itself is correct but it is unnecessary and the interpretation is incorrect. Normalising the MEP to the CMEP accounts for both spinal and peripheral contributions to the MEP, which means the procedure isolates cortical excitability in each participant. This method to evaluate cortical excitability is far superior to a comparison of mean values of MEP and CMEP normalised to Mmax. As such, the interpretation put forth by Sypkes and colleagues (reduced spinal excitability and unchanged cortical excitability) is accurate and the present authors incorrectly state that those data support increased cortical excitability. The second reanalysis conducted by the authors involved pooling the data from two studies (Hahn et al., 2012 and Sypkes et al., 2018) and is even more problematic than the first reanalysis. Measures of neural excitability (i.e., MEP, CMEP, and, to a lesser extent, Mmax) are highly dependent on the conditions of the task, which makes it inappropriate to pool data collected during maximal contractions (Hahn et al., 2012) with data collected at a moderate intensity (Sypkes et al., 2018). Furthermore, the responses can differ greatly between muscles (e.g., Giesebrecht et al., 2010 vs. Gandevia et al., 1999 or Martin et al., 2006 vs. Oya et al., 2008), which makes it inappropriate to pool data from the soleus (Hahn et al., 2012) and TA (Sypkes et al., 2018). Given these critical issues with the reanalysed data, I do not believe the reanalyses have a place in the manuscript. This will have significant implications on the rationale of the study and would require the elimination of Figure 1.

Authors’ response part 1:
We have now removed the reanalysis and Figure 1 from the manuscript. However, this is because we think the reanalysis is problematic because the supplementary material of Sypkes et al. does not provide the M-waves corresponding to the MEPs.

New comment:
Obviously, I approve of the authors’ decision to remove the reanalysis element of the manuscript; however, I am a bit concerned that their sole reason was the absence of M-wave data from the Sypkes study. Even if those data were present, as I pointed out in my original comment, there are fundamental flaws associated with combining data collected at different contraction intensities and from different muscles. I am all in favour of trying to combine similar data to improve statistical power, but the two studies in question are too dissimilar to justify an attempt at merging the datasets.

Authors’ response part 2:
We do not see why the normalization of MEPs to CMEPs should be “far superior” to normalizing both CMEPs and MEPs to their corresponding Mmax. Rather, we actually think that analysing MEPs using CMEPs as the ratio of MEP/CMEP is problematic, unless MEPs and CMEPs are perfectly matched in size so that they can be assumed to represent the exact same motoneuron pool, and, more importantly, when M-waves do not vary between conditions. This is demonstrated in the table below (Table 1, see the reference condition). In the first virtual test condition, MEPs and CMEPs are reduced by the same amount between reference and test conditions (from 10 to 5), which results in a MEP/CMEP ratio of 1 indicating reduced spinal excitability, but unchanged cortical excitability (similar to Sypkes et al. 2018). However, as the Mmax corresponding to the MEPs changed as well (the M-wave MEP became smaller by 20% between reference and test conditions, whereas the M-wave CMEP was unchanged), the normalised MEP/CMEP ratio goes up to 1.25. This virtual finding would subsequently indicate reduced spinal excitability, but increased cortical excitability. In the second virtual test condition, CMEPs are reduced between reference and test conditions (again from 10 to 5), but MEPs are unchanged (similar to Sypkes et al. 2018). While the reduced CMEPs would again indicate reduced spinal excitability, the increased MEP/CMEP ratio of 2 (as expected based on reduced CMEPS, but unchanged MEPs) would indicate increased cortical excitability. However, this calculated ratio might be biased by M-waves varying between conditions.

New comment:
In terms of my comment about normalizing the MEP to the CMEP being far superior to separate comparisons to Mmax, when each of the MEP, CMEP, and Mmax are collected under the same conditions (and the MEP and CMEP are a similar proportion of Mmax size in the control condition, which should always be the aim), there is no benefit to normalizing MEP to Mmax. In this scenario, any change to Mmax, CMEP/Mmax, and MEP/CMEP values indicate changes in peripheral, spinal, and cortical excitability, respectively and nothing of interest comes from MEP/Mmax because corticospinal excitability is less insightful than separate measures of spinal and cortical excitability. In the Sypkes study, the Mmax and MEP were collected in one contraction and the CMEP in another, with the CMEP and MEP at the same time point in each contraction. In retrospect, it would have been best to collect Mmax data from both contractions, which would align with the design portrayed in Table 1 (alternatively one could collect MEP and CMEP in one contraction and CMEP and Mmax in another or all three in a single contraction). However, given the fluctuations in EMG or torque within a contraction (not including the effects caused by different stimuli), there is a case to be made that collection of Mmax and MEP (or Mmax and CMEP) a second or more apart in the same contraction is not necessarily a better comparison than one made across contractions. In the scenario illustrated in Table 1, the Mmax differs by 20% across two contractions of the same type, yet the comparison of MEP/Mmax to CMEP/Mmax assumes (relies on the idea) that spinal excitability is affected similarly in both contractions. I think the authors would agree* that spinal compared to peripheral excitability is far more likely to differ between two like contractions, which means their portrayal of MEP/Mmax vs. CMEP/Mmax as a superior method to MEP/CMEP is misleading. *In their answer to my comment about line 411 of the original manuscript, the authors suggest that they did not collect Mmax because it has been shown not to differ between control and RFE conditions, which means there is even less reason to believe it would differ substantially (e.g., 20%) between two control or two RFE contractions. Ultimately, no scenario allows us to test the different portions of the motor pathway under identical conditions and we have to live with the inherent limitations of any option. Given my concern with the soundness of the argument put forth in the revised Introduction and the associated supplemental Table (see new comment on the revised manuscript), I believe it will only create confusion and mislead readers to include these new elements of the manuscript.

Authors’ response part 3:
Accordingly, when CMEPs are reduced (as in Sypkes et al. 2018), but the MEP/CMEP ratio remains unchanged, the MEPs would need to decrease as well. While this is of course a theoretical consideration, this was not the case in Sypkes et al. 2018, where MEPs were unchanged between conditions. Accordingly, we think that the increased MEP/CMEP ratio from Fig. 3 F in Sypkes et al. (2018) did not achieve significance because the study was underpowered.

New comment:
In terms of absolute size, in most cases, the MEPs did decrease as well, which is why the MEP/CMEP ratio remained unchanged. The authors state “MEPs were unchanged between conditions” but, as absolute MEPs were not provided in the Sypkes article, I believe the authors mean the MEP/CMEP ratio was unchanged (see new comment on lines 108-113 of the revised manuscript). I fully concede that a larger sample size may have led to a statistically significant increase in the MEP/CMEP ratio. That said, if we are working with hypotheticals, it is pretty clear from Figure 3F in the Sypkes study that two participants are responsible for the authors’ perceived trend in the data; i.e., a larger sample size of participants more similar to the rest of group could make the MEP/CMEP line between ISO and RFE even flatter. Who knows?

Authors’ response part 4:
In the revised manuscript, we now show that the previous studies were underpowered, which is why we doubt their findings. On an important note, this is not just related to the Sypkes et al. (2018) study, but also to our own previous study (Hahn et al. 2012). Further, the aim of our study was not to disprove the findings from Sypkes and/or Hahn, but to better understand the neural nature of rFE. Accordingly, we do not care whether corticospinal excitability is changed or unchanged, but we are just interested in valid results and we think reviewer#2 should also be open minded to results not necessarily supporting their own view.

New comment:
I absolutely agree that the studies, like nearly all in human neuromuscular physiology, would benefit from a larger sample size. My original and new comments about the merging of data and advancing the benefits of the MEP/Max vs. CMEP/Mmax comparison have nothing to do with my perceptions of what the data should show (in this case, I have no preconceptions), they were/are based on the validity of the results and arguments put forth by the authors. Like you, I do not care whether an RFE state causes increases or decreases in cortical or spinal excitability. If a study is conducted carefully, the data are what they are.

General comments on the revised manuscript:

Introduction
- lines 105-108: As stated earlier, I think the authors are overstating the challenges of interpreting the MEP/CMEP ratio and overselling the benefits of the MEP/Mmax ratio.
- lines 108-113: I maintain that this interpretation is incorrect because it only makes sense if the data provided in the Sypkes study were absolute CMEPs and MEPs. However, the data provided were normalized not absolute potentials. That is CMEP/Mmax (not CMEP) was reduced and MEP/CMEP (not MEP) was unchanged. As the comparison of MEP/CMEP ratio between conditions represents an isolated measure of cortical excitability, I fail to see how that the authors can reinterpret an unchanged MEP/CMEP ratio to suggest an increase in cortical excitability.

Methods
- lines 323-324: I suggest the authors reword “was surprising assuming no difference”.

---

## Round 0.3 · accepted · Accept

Dear Authors,
Two experts in the field feel that your manuscript could be suitable for publication in the present form. Congratulations.

·

Basic reporting

.

Experimental design

.

Validity of the findings

.

Additional comments

Thank you for allowing me to view the revisions.
My final review was summarized in the previous round
nice work.
Geoff

·

Basic reporting

No comment.

Experimental design

No comment.

Validity of the findings

No comment.

Additional comments

Thank you to the authors for their carefully-considered responses to the last review. Although none of the authors’ arguments won me over to their point of view, I respect their opinions and am happy to conclude the debate. Congratulations.